# UV photochemistry of the L-cystine disulfide bridge in aqueous solution investigated by femtosecond X-ray absorption spectroscopy

Miguel Ochmann[1,6], Jessica Harich [1,6], Rory Ma[2,6], Antonia Freibert [1 ✉], Yujin Kim[2], Madhusudana Gopannagari[3], Da Hye Hong[3], Daewoong Nam [2,4], Sangsoo Kim[2], Minseok Kim[2], Intae Eom [2,4], Jae Hyuk Lee[2,4], Briony A. Yorke [5], Tae Kyu Kim [3 ✉] & Nils Huse [1 ✉]

The photolysis of disulfide bonds is implicated in denaturation of proteins exposed to ultraviolet light. Despite this biological relevance in stabilizing the structure of many proteins, the mechanisms of disulfide photolysis are still contested after decades of research. Herein, we report new insight into the photochemistry of L-cystine in aqueous solution by femtosecond X-ray absorption spectroscopy at the sulfur K-edge. We observe homolytic bond cleavage upon ultraviolet irradiation and the formation of thiyl radicals as the single primary photoproduct. Ultrafast thiyl decay due to geminate recombination proceeds at a quantum yield of >80 % within 20 ps. These dynamics coincide with the emergence of a secondary product, attributed to the generation of perthiyl radicals. From these findings, we suggest a mechanism of perthiyl radical generation from a vibrationally excited parent molecule that asymmetrically fragments along a carbon-sulfur bond. Our results point toward a dynamic photostability of the disulfide bridge in condensed-phase.

The disulfide bond moiety plays a key role in the thermal stability of proteins: During protein folding, two L-cysteinyl residues in the amino acid chain can be covalently coupled to form a disulfide bond[1], either intramolecularly within the same protein monomer[2] or intermolecularly between two protein monomers[3]. This disulfide cross-linkage thus stabilizes the tertiary and quaternary structure of proteins. Furthermore, the disulfide bond can act as a radical scavenger, protecting the protein from damage by reactive oxygen and nitrogen species (RONS)[4]. Such reactive species may be generated upon exposure to ultraviolet (UV) radiation or in response to oxidative stress resulting in redox imbalance[5]. RONS can permanently damage the protein through amino acid oxidation and peptide cleavage, resulting in loss of function and potentially harmful aggregation. In addition,

free thiol groups and thioethers from L-cysteine and L-methionine are implicated in radical damage and repair mechanisms[6,7].

The disulfide UV photochemistry has been the topic of many studies which have focused on the reactions that may result when breaking S-S or C-S bonds and forming the respective thiyl or perthiyl radicals[8]. The picture emerging from the sum of static and time-resolved studies conducted over a time span of nearly 75 years is one of a clear distinction between reactions in the gas phase and in solution.

Static and time-resolved studies in the gas phase established that excitation at wavelengths shorter than 200 nm can break both the C-S and the S-S bond in dimethyl disulfide (DMDS)[9–13]. Bookwalter et al. showed the same to be true for low-order di-n-alkyl disulfides[14]. With

[1]Department of Physics, University of Hamburg and Center for Free-Electron Laser Science, Hamburg, Germany. [2]Pohang Accelerator Laboratory, POSTECH, Pohang, Republic of Korea. [3]Department of Chemistry, Korea Advanced Institute of Science and Technology (KAIST), Daejeon, Republic of Korea. [4]Photon Science Center, Pohang University of Science and Technology, Pohang, Republic of Korea. [5]School of Chemistry, University of Leeds, Leeds, United Kingdom. [6]These authors contributed equally: Miguel Ochmann, Jessica Harich, Rory Ma. ✉e-mail: antonia.freibert@physik.uni-hamburg.de; taekyu.kim@kaist.ac.kr; nils.huse@uni-hamburg.de

the exception of Rinker et al.[15], excitation at wavelengths longer than 200 nm of various disulfides in the gas phase exclusively leads to thiyl radical formation ranging from the smallest disulfide DMDS to disulfide-containing proteins[13,14,16–23]. A theoretical study by Luo et al. supports these experimental findings in that population of the two lowest electronically excited states leads to dissociation of the S-S bond[24]. Femtosecond studies of disulfides in the gas-phase have been performed by Stephansen et al.[19,20] (1,2-dithiane) as well as Schnorr et al.[23] (DMDS), both reporting the exclusive formation of thiyl radicals. While the latter study established a thiyl radical formation time of (120 ± 30) fs, Stephansen et al. showed that the bond between the two sulfur atoms (which were connected by a carbon chain) reforms with a time-constant of (2.75 ± 0.23) ps, leading them to suggest an inherent photostability of disulfide bridges in confined spaces.

In solution the photoreactions of disulfides become more complicated. The fission of the S-S bond has also been reported upon irradiation with wavelengths greater than 200 nm[8,25–33]. Nevertheless,

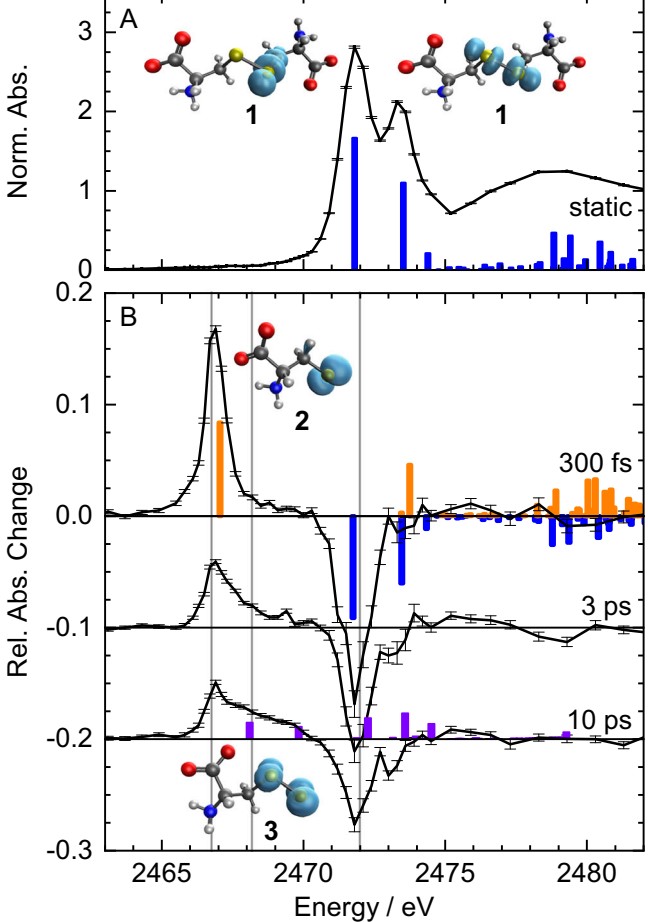

**Fig. 1 | Static and differential sulfur K-edge absorption spectra of L-cystine with TD-DFT-calculated transitions.** Error bars represent standard deviations of the 600 measurement values acquired in 10 s from which each data point was calculated. **A** Static spectrum of L-cystine (**1**, black) with calculated lowest vertical excitation energies (blue sticks) and corresponding isosurface plots of the difference electron attachment densities (turquoise). **B** Differential spectra taken 0.3 ps, 3 ps and 10 ps after 267-nm excitation (black). Calculated transitions for the parent ion are shown in blue, for the thiyl radical (**2**) in orange and for the perthiyl radical (**3**) in purple. Isosurface plots (turquoise) show the beta attachment densities of the dominant excitation. The absorption changes are normalized to the K-edge height of L-cystine, i.e. the thiyl radical peak absorption at 300 fs amounts to 17 % of the parent molecule's K-edge jump. Vertical gray lines indicate the energies at which delay scans were recorded (c.f. Figure 3). Source data are provided as a Source Data file.

several studies observe photoproducts as a result of C-S bond scission[34–41]. Notably, the latter observations were reported predominantly on tertiary disulfides such as *tert*-butyl disulfide and penicillamine disulfide. However, the temporal resolution of hundreds of nanoseconds or longer does not allow for unequivocally establishing a reaction mechanism.

We have previously demonstrated that time-resolved X-ray absorption spectroscopy (TRXAS) at the sulfur K-edge is a chemically specific tool to observe the UV photochemistry of organosulfur compounds in solution[42,43], with the observation that both C-S and S-S bond cleavage are readily accessible at 267 nm excitation within 70 ps, and that the complete decay of both photoproducts, the thiyl and the perthiyl radicals, occurs on timescales from 0.1 ns up to 150 ns[43]. Previous observations of apparently longer living perthiyl radicals formed from *tert*-disulfides are potentially due to stabilization of sulfur radicals by tertiary carbon substituents attached to the disulfide moiety, significantly prolonging radical life times into the range of microseconds[27,35,39].

The contradicting results between the isolated molecules and molecules in a condensed phase clearly indicate that further investigation is needed to characterize the role of the disulfide environment in product formation and to clarify the conditions necessary to access a specific photochemical reaction pathway. Reactions subsequent to initial fragmentation such as geminate recombination are in most cases inaccessible in the gas-phase but common in solution[17,44–46] due to the confining forces of a solvent cage or a protein backbone[47,48]. This aspect is of particular importance when considering the role of disulfide photochemistry with respect to protein photostability and function. Previous attempts have been made to mimic disulfide photochemistry occurring in solution or in proteins by investigating cyclic disulfides in which the terminal biradical geminates are connected by a hydrocarbon chain[19,49,50]. However, several processes that involve an environment acting as a thermodynamic bath and allowing for energy dissipation as well as confinement and escape, are crucially determining factors for disulfide chemistry.

In this study, we investigate the photodissociation dynamics of L-cystine, the disulfide dianion of the amino acid L-cysteine, in aqueous solution upon pulsed UV irradiation with 267 nm light using TRXAS at the sulfur K-edge in the tender X-ray regime (2-5 keV)[51–53]. We observe the disulfide photochemistry on timescales of 100 fs up to hundreds of picoseconds in aqueous solution, thereby clearly identifying the primary and subsequent reaction steps that allow to establish chemical cycles of light-induced disulfide bond cleavage.

## Results and discussion
### X-ray absorption spectra
The static X-ray absorption spectrum of L-cystine (**1**) at the sulfur K-edge is shown in Fig. 1A. It exhibits two clearly discernible transitions near 2472 eV that are spaced by about 1.5 eV. At higher energy the beginning of the extended X-ray absorption fine-structure (EXAFS) is visible. Note that we have normalized the absorption spectrum to the sulfur K-edge around which the EXAFS modulation oscillates, as is customary in X-ray spectroscopy. In Fig. 1B differential X-ray absorption spectra are plotted for time delays of 0.3 ps, 3 ps and 10 ps between 267-nm excitation pulses and X-ray probe pulses. Negative signals indicate a loss of absorption (sample bleaching) while positive signals mean increased absorption.

At a time delay of 300 fs, a new absorption lineshape has appeared at 2466.9 eV and bleaching can be observed in the region of the lowest sulfur-1s absorption lines of L-cystine. At 3 ps time delay the differential spectrum has partially relaxed to smaller levels. Most distinctly, a shoulder has emerged on the higher energy side of the induced absorption lineshape and the bleach signal exhibits two pronounced minima at energies that coincide with the absorption peaks in Fig. 1A. The broad induced absorption shoulder has fully emerged 10 ps after

excitation and the bleach signal clearly resembles a negative replica of the lowest sulfur-1s transitions of L-cystine.

To interpret the spectra in Fig. 1, we have modeled X-ray absorption lines of L-cystine and possible chemical products by time-dependent density functional theory (TD-DFT) and also benchmarked these computational results with results from the algebraic diagrammatic construction scheme (ADC2-x). Additional computational information is provided in the Supplementary Information. The lowest calculated sulfur-1s transitions of L-cystine coincide very well with the experimental transitions in Fig. 1A. We identify these transitions as the excitation of sulfur-1s electrons to sulfur-based π*-orbitals at positions characteristic for aliphatic disulfide bonds. The induced absorption lineshape at 2466.9 eV in Fig. 1B is well described by the lowest sulfur-1s transition of the L-cysteinylthiyl radical (**2**) which has a second transition that overlaps with the second absorption peak of L-cystine (explaining the near lack of a bleach signal at 2473.5 eV). This excellent agreement between theory and experiment identifies the primary (and only) photoproduct as the two identical thiyl radicals produced from homolytic S−S bond cleavage of L-cystine within 300 fs after 267-nm excitation. This finding establishes that the initial reaction pathways in the gas-phase and in solution are identical. We note that our results are fully consistent with those of Schnorr and coworkers who also established a threshold for two-photon absorption (20 mJ/cm² @ 30-fs excitation)[23] in DMDS that we find for aqueous L-cystine as well (75 mJ/cm² @ 100-fs excitation). At higher peak-power density we observe a broad shoulder between the lowest thiyl transition and the bleach signal within the time resolution of the experiment.

The lineshape of the thiyl radical is well described by a Lorentzian because the energy resolution of the experiment is well below the natural linewidth of sulfur-1s absorption lines. Interestingly, the L-cystine lineshapes in Fig. 1A are inhomogeneously broadened (see lineshape analysis in Supplementary Fig. 1) which we attribute to a distribution of L-cystine conformers that are thermally accessible at room temperature in solution.

The change in differential absorption at time delays of 3 ps and 10 ps clearly indicates a more complex disulfide photochemistry in solution. A secondary product manifests in the form of a broad absorptive shoulder that rises at 2468 eV and higher energy, between

the lowest thiyl radical transition and the bleach signal. The time delay of 3 ps precludes a diffusive process as the underlying cause for secondary product formation. Thiyl radicals also do not react with water molecules[4] and we can rule out significant formation of solvated electrons in the vicinity of thiyl radicals at our experimental peak-power densities as outlined in the Supplementary Note 6. We therefore consider recombination of the geminate thiyl radicals as the underlying cause for secondary product formation which we identify as the L-cysteinylperthiyl radical (**3**). Firstly, theory provides the best match between the lowest absorptive transitions of perthiyl radicals and additional experimental absorption changes at picosecond time delays. Secondly, the bond dissociation energy (BDE) of the carbon-sulfur bond is about 20 % lower than the one of the S-S bond[54,55], making this reaction pathway energetically accessible. Lastly, other chemical species such as the lowest triplet state of L-cystine (in which the sulfur lone-pair orbitals are aligned) or an L-cysteine anion-cation pair are energetically much less likely as we detail in the Supplementary Note 1 on potential reaction products.

The proposed secondary reaction pathway upon geminate recombination is illustrated in Fig. 2. UV excitation at 267 nm wavelength (4.6 eV) populates an antibonding state in disulfides (vertical arrow) which must feature an energy barrier of 6.2 eV (200 nm wavelength) along the C-S bond because only S-S bond cleavage has been reported in the literature for excitation energies below 6.2 eV. The thiyl radicals are formed at a bond energy of 2.9 eV[54], the minimum energy at which geminate recombination occurs with respect to the ground-state energy of L-cystine, thereby providing energy in access of the C-S bond dissociation energy of 2.3 eV[55]. If the time-scale of C-S bond cleavage is on the order of or shorter than the relaxation of a recombined parent molecule which is naturally in a highly vibrationally excited state (VES), perthiyl formation is energetically well possible.

## Temporal product evolution

The spectral evolution of the differential sulfur K-edge absorption already reveals the essentials of the earliest L-cystine photochemistry in aqueous solution. However, we can quantify chemical reaction rates and yields by following the time evolution of the absorption changes at characteristic spectral positions. We probed the absorption changes, which are proportional to the concentration of the primary and secondary photoproducts as well as the loss of the parent compound, at the three energies indicated in Fig. 1 by gray vertical lines.

The transients are plotted in Fig. 3. The bleach signal at 2472.0 eV (blue) emerges within our time resolution of 200 fs and decays on multiple timescales. About a third of the signal vanishes within the first 2 ps, and about a third of the signal persists much longer than the largest time delay of 0.8 ns. In contrast, more than half of the induced absorption signal at 2466.8 eV (orange) decays within the first 2 ps. Further signal decay occurs on tens of picoseconds. At a time delay of 20 ps less than one fifth of the signal remains and appears to persist beyond 0.8 ns. The induced absorption that we associate with the secondary photoproduct was probed at 2468.2 eV (purple). The initial signal rises within our time resolution and continues to grow over tens of picoseconds before it partially decays on hundreds of picoseconds with a persistent signal at the longest time delay of 0.8 ns. The inset of Fig. 3 shows a temporal zoom of the time scan at 2468.2 eV between −0.5 ps and +1.5 ps along with the optimal fit of the kinetic model (light purple) and the two modeled contributions (orange and purple) to the signal at this energy.

We note three key observations that motivate our kinetic model: Firstly, we observe a signal relaxation at the energy positions that we associate with the primary photoproduct and parent molecule bleaching within the first 2 ps and the first 20 ps. Secondly, the signal at the probe energy associated with the secondary photoproduct rises on these two timescales. Thirdly, the signal at 2466.8 eV appears to be

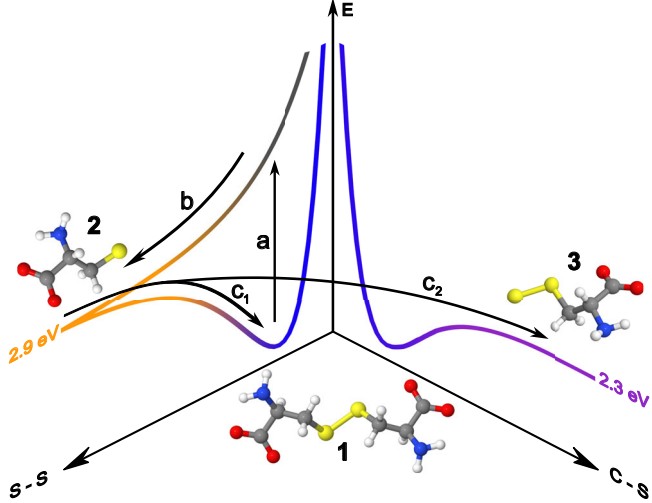

**Fig. 2 | Schematic of the proposed initial and secondary reaction pathways.**
After photoexcitation (a), the L-cystine molecule (**1**, blue) splits homolytically along the S-S coordinate (b) into two identical thiyl radicals (**2**, orange). Geminate recombination of these thiyl radicals occurs at a BDE$_{S-S}$ = 2.9 eV[54], energetically high enough for C-S bond cleavage (c₂) at a BDE$_{C-S}$ = 2.3 eV[55], yielding a perthiyl radical (**3**, purple). Alternatively, the parent ground state molecule **1** is reformed by intramolecular energy relaxation (c₁).

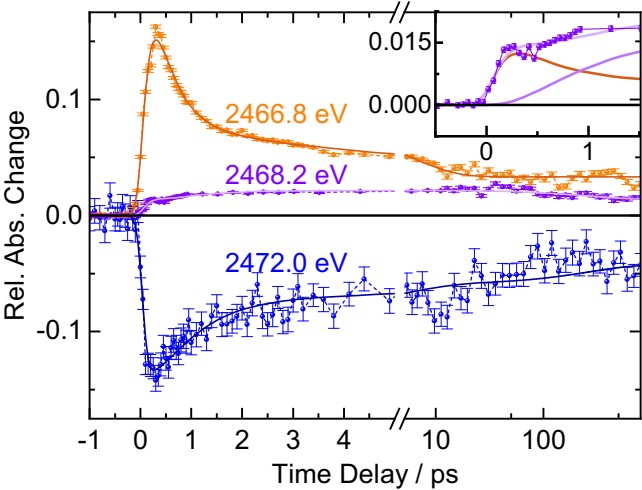

**Fig. 3 | Delay scans of the relative absorption change at indicated energies along with model fits.** The rate-equation model is illustrated in Fig. 4. Error bars represent standard deviations of the 600 measurement values acquired in 10 s from which each data point was calculated. Slower variations of experimental conditions lead to additional noise. The scale of the x-axis changes from linear to logarithmic after the axis break at 5 ps with minor tick marks at 50 ps and 500 ps. The induced absorption at 2466.8 eV we associate with the L-cysteinylthiyl radical (**2**, orange). The induced absorption at 2468.2 eV is attributed to the L-cysteinylperthiyl radical (**3**, purple). The bleach signal at 2472.0 eV follows the transient loss of L-cystine (**1**, blue). The inset shows the first 1.5 ps of the delay scan at 2468.2 eV. At this energy, the absorption lineshape of the thiyl radical contributes with 8 % of its maximum value to the signal. We therefore modeled the delay scan at 2468.2 eV as the sum (light purple curve in inset) of the secondary photoproduct (purple curve in inset) and 8 % of the thiyl absorption at 2466.8 eV (orange curve in inset). See Supplementary Notes 1 and 4 for details. Source data are provided as a Source Data file.

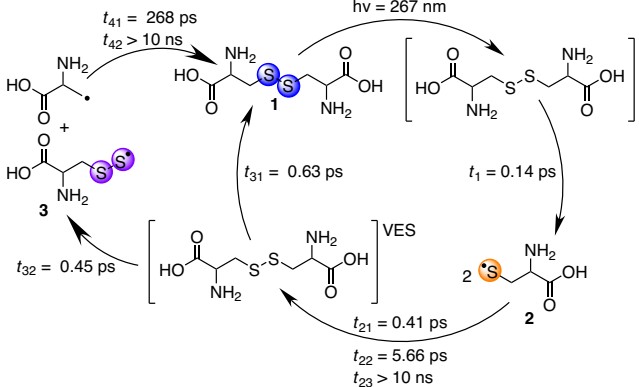

**Fig. 4 | Photochemical cycles proposed for UV-excited L-cystine (1, blue) in aqueous solution.** Upon illumination with 267-nm light, the S-S bond in **1** is cleaved homolytically yielding two L-cysteinylthiyl radicals (**2**, orange). Geminate recombination facilitates C-S bond cleavage to form the L-cysteinylperthiyl radical (**3**, purple) as a secondary reaction from the parent in a vibrationally excited state (VES). Alternatively, **1** is reformed by intramolecular energy relaxation or by recombination of the perthiyl and carbonyl radicals.

constant between 20 ps and 0.8 ns while the transients at 2468.2 eV and at 2472.0 eV decay slightly in this time interval.

We have fit a rate-equation model to the data in Fig. 3. The model is described in detail in the Supplementary Note 4. Importantly, the multiple timescales suggest that at least three different sub-ensembles exist that can be associated with the three timescales on which we see signal changes, i.e. 2 ps, 20 ps and much longer timescales. In this

**Table 1 | Optimized global fit parameters of the kinetic model employed**

| time constants | relative yields |
|---|---|
| $\tau_1/ps = 0.14 \pm 0.02$ | 1 |
| $\tau_{21}/ps = 0.41 \pm 0.05$ | $q_{21} = 0.67 \pm 0.02$ |
| $\tau_{22}/ps = 5.66 \pm 0.51$ | $q_{22} = 0.19 \pm 0.01$ |
| $\tau_{31}/ps = 0.63 \pm 0.26$ | $q_{31} = 0.66 \pm 0.04$ |
| $\tau_{32}/ps = 0.45 \pm 0.06$ | $1 - q_{31}$ |
| $\tau_{41}/ps = 268 \pm 165$ | $q_{41} = 0.42 \pm 0.11$ |

model, thiyl radicals either geminately recombine within 2 ps, within 20 ps, or not at all within our observation window of 0.8 ns. This model can be rationalized physically in the following way: The majority of thiyl radicals geminately recombines very rapidly within 2 ps. Because of the fluctuating solvent environment some thiyl radical pairs will rotate substantially during dissociation which will require rotational diffusion for geminate recombination, the timescale of which aligns very well with the 20-ps range we observe. Some fraction of radical pairs escapes the solvent cage and their recombination will take place on diffusive timescales that are beyond the sub-nanosecond time range that we have probed.

We have summarized the result of our kinetic modeling in Fig. 4 with exponential time constants for radical formation ($\tau_1$), geminate recombination ($\tau_{2i}$), energy relaxation/perthiyl formation ($\tau_{3i}$), and C-S bond reformation ($\tau_{4i}$). We find a thiyl radical formation time of (140 ± 50) fs which aligns very well with the 120 fs that Schnorr and co-workers find for gas-phase DMDS[23]. Notably, dominant S-S bond reformation with a quantum yield of $q_{21} = 0.67$ is ultrafast with a time-constant of $\tau_{21} = 410$ fs, making it an order of magnitude faster than what gas-phase experiments have observed for 1,2-dithiane[19,20]. The ultrafast geminate thiyl recombination highlights the importance of confinement and energy dissipation in condensed phases. The additional decay of the thiyl radical signal manifests in a second decay constant of $\tau_{22} = 5.7$ ps at a quantum yield of $q_{41} = 0.19$. This means that five out of six thiyl radical pairs recombine within 20 ps.

Our model also provides a relative yield of $q_{32} = 0.34$ for secondary product formation, i.e. every third recombination of two thiyl radicals leads to C-S bond cleavage. Almost half of the perthiyl radicals recombine in less than a nanosecond (with a relative yield of $q_{41} = 0.42$ and a time constant of $\tau_{41} = 270$ ps). In sum, 70 % of dissociated L-cystine molecules have recombined within 1 ns. Because our maximum time delay is 0.8 ns, we estimate that longer-lived radical products must have time constants larger than 10 ns. Table 1 contains the relative yields and time constants extracted from fitting the kinetic model to the delay scans in Fig. 3.

The high yield with which the S-S bond in aliphatic disulfides reforms on ultrafast timescales upon homolytic bond cleavage suggests that the charge density of the thiyl radicals does not relax substantially but allows for facile bond formation and vibrational energy dissipation such that the high-energy quantum of the UV photon, which was initially absorbed, can be dissipated effectively by bond cleavage and geminate recombination. The photocycles of disulfide chemistry in the condensed phase may therefore have little consequence for the structural integrity of disulfide-stabilized proteins, due to a dynamical photostability of the aliphatic disulfide bond motif.

## Concluding remarks

We have elucidated the UV photochemistry of L-cystine upon 267-nm excitation by employing femtosecond X-ray absorption spectroscopy at the sulfur K-edge. Solvent-mediated primary geminate recombination occurs effectively via a vibrationally highly excited state, which can lead to perthiyl radical formation, albeit in minority yield. This observation underscores the very effective radical quenching ability of

aliphatic thiyl radicals by recombination and the importance of sulfur functional groups in radical quenching reactions.

More generally, X-ray absorption spectroscopy at the sulfur K-edge is a valuable tool for gaining information about the UV photochemistry of organosulfur compounds, in the condensed phase, suggesting applications in biologically relevant settings, for instance in proteins, where backbone motion plays a role for the recombination probability of thiyl radicals[56]. In summary, we clarified that in solution only homolytic cleavage of the disulfide bond occurs upon UV excitation below 6.2 eV (>200 nm wavelength), generating exclusively thiyl radicals as primary photoproducts. Furthermore, we suggest a mechanism for the generation of perthiyl radicals in solution at relatively low yield in a secondary reaction step, thereby providing a possible answer to a long-standing discussion. It will be very interesting to follow disulfide photochemistry in proteins where, in conjunction with aromatic residues, additional reaction pathways such as charge and energy transfer will influence photo-driven disulfide chemistry.

## Methods

### Experimental setup

All measurements were conducted at the NCI beamline of the PAL-XFEL free electron laser, Republic of Korea. The setup has been described in detail elsewhere[57–59]. Briefly, the sample was excited with the third harmonic (267 nm) of a Ti:Sa amplified laser system at 30 Hz repetition rate and 100 fs pulse duration (FWHM) of the UV pulses. The round laser focus size was 145 μm ($1/e^2$). For differential XAS spectra and transients a power density of 0.75 TW cm$^{-2}$ was used. Assuming an X-ray pulse width of $\sigma_{x-ray} = 20$ fs, a laser pulse width of $\sigma_{laser} = 45$ fs, a broadening from the sample due to group velocity mismatch of 65 fs over twice the X-ray attenuation length and a temporal jitter/drift between X-rays and laser of 30 fs, the total expected width of the instrument response function (IRF) amounts to $\sigma_{IRF} = 87$ fs. The X-rays were delivered in a pulse train of 60 Hz and X-ray absorption and changes thereof were detected in fluorescence-yield mode by an avalanche photodiode (APD) shielded with an aluminum foil (thickness: 1.6 μm). The X-rays are focused to $14 \times 15$ μm$^2$ at the sample position. The energy axis o our measurements was shifted by + 0.3 eV according to our calibration, which takes several X-ray sources into account (see Supporting Information of Ochmann et al.[43]). The sample was delivered through a round steel nozzle of 150 μm diameter into the sample chamber that was filled with helium at ambient pressure.

### Materials

L-cystine (99.7 % TLC) and sodium hydroxide (BioXtra, ≥98 %) were purchased from Sigma-Aldrich Republic of Korea and were used as received without further purification. In total 200 mL of an aqueous solution containing 4.80 g L-cystine (100 mM) and 2.39 g NaOH (300 mM) were prepared. Half of the sample solution was loaded into the jet's sample reservoir for measurements. After 5 h of running in a continuous loop, the sample was replaced with the remaining fresh sample solution. No change in X-ray absorption of the unexcited sample was observed during the measurement intervals.

### Kinetic modeling

We have modeled transient delay scans at three characteristic spectral positions by the solution of a rate-equation model convolved with Gaussian instrument response function, where the $N_{ij}(t)$ are the time-dependent partial populations with $N_1$ as the excited-state population, $N_2$ as the population of **2**, $N_3$ as the population of the vibrationally excited parent molecule, $N_4$ as the population of **3** and $N_0$ as the population of **1** at thermal equilibrium. The populations of $N_2$, $N_3$, and $N_4$ are a sum of sub-ensembles to account for multiple relaxation timescales as observed in the experiment. Here, $k_1 = 1/\tau_1$

and $k_{ij} = 1/\tau_{ij}$ are the rate constants of the corresponding population decay and $q_{ij}$ are the relative yields of a sub-ensemble:

$$\dot{N}_1(t) = -k_1 N_1(t) \tag{1}$$

$$\dot{N}_{2i}(t) = k_1 N_1(t) - k_{2i} N_{2i}(t) \tag{2}$$

$$\dot{N}_{3j}^{2i}(t) = k_{2i} N_{2i}(t) - k_{3j} N_{3j}^{2i}(t) \tag{3}$$

$$\dot{N}_{4j}^{2i}(t) = k_{32} N_{32}^{2i}(t) - k_{4j} N_{4j}^{2i}(t) \tag{4}$$

$$\dot{N}_{031}^{2i}(t) = k_{31} N_{31}^{2i}(t), \dot{N}_{04j}^{2i}(t) = k_{4j} N_{4j}^{2i}(t) \tag{5}$$

$$N_2 = \sum_{i=1}^{3} q_{2i} N_{2i} \tag{6}$$

$$N_3 = \sum_{j=1}^{2} q_{3j} N_{3j} = \sum_{j=1}^{2} q_{3j} \sum_{i=1}^{3} q_{2i} N_{3j}^{2i} \tag{7}$$

$$N_4 = \sum_{j=1}^{2} q_{4j} N_{4j} = \sum_{j=1}^{2} q_{4j} \sum_{i=1}^{3} q_{32} q_{2i} N_{4j}^{2i} \tag{8}$$

$$N_0 = q_{31} \sum_{i=1}^{3} q_{2i} N_{031}^{2i} + \sum_{j=1}^{2} q_{4j} \sum_{i=1}^{3} q_{32} q_{2i} N_{04j}^{2i} \tag{9}$$

and $N_1(0) = 1, N_2(0) = N_3(0) = N_4(0) = N_0(0) = 0$.

In the Supplementary Discussion of the kinetic model the solution for the differential equation system is given and a schematic representation of the proposed reaction scheme is shown in the Supplementary Fig. 3. Within this model the first state $N_1$ and the vibrationally excited parent $N_3$ are not observed with our experimental conditions. $N_1$ is assumed to feed into the first state observed, $N_2$. $N_2$ then feeds into the state $N_3$ that does not accumulate observable population. The modeled width $\sigma_{IRF}$ of the instrument response function (IRF), assuming a Gaussian shape, was fit as $\sigma_{IRF} = (0.087 \pm 0.006)$ ps, giving an experimental time resolution of about 200 fs full width at half maximum (FWHM). The time delay scans were modeled with global time constants $\tau_1$ and $\tau_{ij}$ by fitting them simultaneously to all three transients.

### Theory

Equilibrium structure optimizations of all molecular species were performed with second-order Møller-Plesset perturbation theory[60] (MP2) using the Dunning correlation-consistent basis set[61] aug-cc-pvtz, and vibrational frequency calculations confirmed the finding of true energetic minima. The atomic coordinates of all optimized structures are provided as Supplementary Dataset 1. Ground-state calculations were carried out using the quantum chemistry software package Gaussian[62].

The X-ray absorption transitions were simulated using time-dependent density functional theory (TD-DFT) at the level of the PBE0/def2-TZVP(-f) functional[63,64] using the RIJCOSX approximation. The conductor-like polarizable continuum model using water as the solvent model was applied, too. The transition energies of all calculated species were shifted by 52.09 eV to match the experimental spectra. All X-ray transition properties were calculated using the quantum chemistry program Orca[65,66].

## Data availability
Source data are provided with this paper.

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

## Acknowledgements

The experiments were carried out using the FXL and CXI instrument at PAL-XFEL (proposal No. 2019-2nd-NCI-032) funded by the Ministry of Science and ICT of Korea. This work was supported by the Global Science Experimental Data Hub Center (GSDC) for data computing and the Korea Research Environment Open NETwork (KREONET) for network service provided by the Korea Institute of Science and Technology Information (KISTI). Supported by the National Research Foundation of Korea, grant No. 2022M3H4A1A04074153: R.M., J.H.L., D.N. and I.E. Funding from the National Research Foundation of Korea, 2022R1A2C3003081: Y.K., M.G., D.H.H., and T.K.K. Funding from the Samsung Science & Technology Foundation fund by Samsung Electronics, SSTF-BA2401-04: Y.K., M.G., D.H.H., and T.K.K. Financial support from the Cluster of Excellence 'CUI: Advanced Imaging of Matter' of the Deutsche Forschungsgemeinschaft (DFG) - EXC 2056 - project ID 390715994: J.H., N.H. and A.F. Funding from the International Max Planck Graduate School for Ultrafast imaging & Structural Dynamics (IMPRS-UFAST) and from the Christiane-Nüsslein-Vollhard-Foundation: A.F.

## Author contributions

M.O. and N.H. conceived and planned the project. M.O., J.H. and N.H. planned and coordinated the experiments. J.H., R.M., J.H.L., D.N., S.K., I.E., M.K., Y.K., M.G., D.H.H., T.K.K., N.H. executed the experiments. J.H. and N.H. analyzed the experimental data. A.F. performed the theoretical calculations. M.O., J.H., B.Y. and N.H. wrote the manuscript with input from all authors.

## Funding

## Competing interests

The authors declare no competing interests.
