## [Peer Review File · Nature Communications]

UV photochemistry of the L-cystine disulfide bridge in aqueous solution investigated by femtosecond X-ray absorption spectroscopyReviewer #1 (Remarks to the Author):

This article is a neat example of the uniqueness of ultrafast X-ray absorption spectroscopy at XFELs in elucidating a photochemical reaction, which is nearly impossible to investigate by optical domain spectroscopy. Indeed, the system in question (Cystine) has high lying electronic states and it would be almost impossible to probe their evolution in an unambiguous way by optical spectroscopy.

On the other hand, the case in question is elegantly probed by X-ray spectroscopy and the data reported in this paper speak for themselves. The addition of theory only comes as a confirmation of the analysis of experimental data.

The paper is clearly written and I have no objections to raise. I just have one point: the geminate recombination time of 2 ps is rather long for such a process and may suggest an interplay with the solvent molecules, were it not for the fact that this time scales is close to the 2.75 ps reported by Stephansen et al in their gas phase study. Can the authors comment on this point?

Reviewer #2 (Remarks to the Author):

This article studies the photorelaxation of L-cystine as a model of e.g. disulfide bridge in proteins. The main novelty of the study is the use of time-resolved X-ray to follow the process in solution, as opposed to gas phase performed in previous studies, therefore being a better model of biological processes. Using their experimental results, supported by some theoretical calculations, the authors suggest a mechanism of the photorelaxation and can clearly identify the impact of the solvent on the resulting timescales. This makes it a valuable addition to the previous research.

While my main expertise lie in the theoretical side, both the experimental and theoretical results seem of good quality and support well their reported conclusions. I believe the article can be published as is. I only have minor comments regarding some of the results in SI.

The SI reports calculations on the triplet state of L-cystine, which are not at all discussed either in the SI or the main manuscript. The crossing to the triplet state is a possible event, and so it would be natural to discuss if the simulated signature is found in the experimental spectra. Similarly, in addition to TD-DFT, the authors report ADC(2)-x results in the SI but they are again not discussed or referred to in the SI and manuscript. I appreciate the presence of these results as validation, and believe a sentence in the manuscript or at least the SI should comment on them.

Reviewer #3 (Remarks to the Author):

We are pleased to provide our review of the manuscript titled "UV photochemistry of the L-cystine disulfide bridge in aqueous solution investigated by femtosecond X-ray absorption spectroscopy".

The manuscript uses femtosecond X-ray absorption spectroscopy to investigate the UV photochemistry of the L-cystine disulfide bridge in aqueous solution. The authors report the homolytic cleavage of the disulfide bond upon UV irradiation, resulting in the formation of thiyl radicals as primary photoproducts. These radicals exhibit ultrafast decay, primarily due to

geminate recombination, forming perthiyl radicals as secondary products. The study provides insights into the dynamic photostability of the disulfide bond and the mechanism of perthiyl radical formation, contributing to the broader understanding of disulfide photochemistry in biological contexts. We find that the investigated topic is of high relevance for the understanding of protein stability and that the study finds a novel understanding of the relevant photochemistry. We therefore recommend the manuscript for publication in Nature Communications with minor revisions as detailed below:

1. "Thiyl radicals also do not react with water molecules and we can rule out significant formation of solvated electrons in the vicinity of thiyl radicals at our experimental peak-power densities." Please add a citation for this statement, as this cannot be expected to be commonly known by the broad readership of NCOMMS.
2. Figure 3: The insert in Figure 3 requires a revision and a more comprehensive explanation. The insert shows delay scans at specific spectral positions associated with different photoproducts and transient loss signals. We have had some difficulty in understanding the relations between the overall figure and the inset, perhaps due to a wrong choice of colour scheme for the curves? Perhaps it just requires a better explanation, but we were not able to connect the inset with the main figure.
3. Orange and Blue "Mirror Discrepancy" in Figure 3: The qualitative description that the photoproduct absorption is qualitatively mirroring the bleach signal needs clarification. It appears to refer to the observation that the absorption signal at 2466.8 eV (orange) and the bleach signal at 2472.0 eV (blue) supposedly exhibits an inverse relationship. However, this does not appear to be exactly the case at high time delays, where the photoproduct signal is mostly constant, and the bleach signal is still decaying. Please provide an explanation. A more precise explanation of this behavior in the context of the kinetic model would be beneficial.
4. Also at long time delays, the fit to the photoproduct signal (orange) at 2466.8 eV requires some explanation. What is the cause of the sudden dip at 10 ps (definitely after the axis time gap)? And, given the small error bars, it seems worth discussing if there is a slow oscillation or perturbation at long time scales.

Overall, the manuscript presents significant findings in the field of UV photochemistry of disulfide bonds. The experimental approach and data analysis are robust, and the conclusions drawn are supported by the presented evidence. However, enhancing the clarity of figure 3 and providing more detailed explanations of the kinetic models will strengthen the manuscript. We hope these suggestions will be helpful for the authors in revising their manuscript. Thank you for the opportunity to review this interesting study.

Reviewer #4 (Remarks to the Author):

Responses to the comments from the Reviewer 1

Reviewer's comment:

This article is a neat example of the uniqueness of ultrafast X-ray absorption spectroscopy at XFELs in elucidating a photochemical reaction, which is nearly impossible to investigate by optical domain spectroscopy. Indeed, the system in question (Cystine) has high lying electronic states and it would be almost impossible to probe their evolution in an unambiguous way by optical spectroscopy.

On the other hand, the case in question is elegantly probed by X-ray spectroscopy and the data reported in this paper speak for themselves. The addition of theory only comes as a confirmation of the analysis of experimental data.

The paper is clearly written and I have no objections to raise. I just have one point: the geminate recombination time of 2 ps is rather long for such a process and may suggest an interplay with the solvent molecules, were it not for the fact that this time scales is close to the 2.75 ps reported by Stephansen et al in their gas phase study. Can the authors comment on this point?

Reply:

We thank the reviewer for the positive feedback. With regard to the point the reviewer raises, we agree that solvent interaction is crucial and we attribute the picosecond time constant to a frictional interaction with the solvent environment that is related to rotational diffusion of rotationally misaligned geminate thiyl radicals before recombination can occur. Indeed, the time constant is too short for translational diffusion or secondary recombination after separation of the geminates by solvent molecule insertion. The similarity in timescales to the study of Stephansen and co-workers may be incidental because in the gas-phase, intramolecular energy redistribution should dictate the timescale of S-S bond reformation. We have expanded the discussion of the time constants in the manuscript on page 6.

Responses to the comments from the Reviewer 2

Reviewer's comment:

This article studies the photorelaxation of L-cystine as a model of e.g. disulfide bridge in proteins. The main novelty of the study is the use of time-resolved X-ray to follow the process in solution, as opposed to gas phase performed in previous studies, therefore being a better model of biological processes. Using their experimental results, supported by some theoretical calculations, the authors suggest a mechanism of the photorelaxation and can clearly identify the impact of the solvent on the resulting timescales. This makes it a valuable addition to the previous research.

While my main expertise lie in the theoretical side, both the experimental and theoretical results seem of good quality and support well their reported conclusions. I believe the article can be published as is. I only have minor comments regarding some of the results in SI.

The SI reports calculations on the triplet state of L-cystine, which are not at all discussed either in the SI or the main manuscript. The crossing to the triplet state is a possible event, and so it

would be natural to discuss if the simulated signature is found in the experimental spectra. Similarly, in addition to TD-DFT, the authors report ADC(2)-x results in the SI but they are again not discussed or referred to in the SI and manuscript. I appreciate the presence of these results as validation, and believe a sentence in the manuscript or at least the SI should comment on them.

Reply:

We also thank reviewer 2 for taking the time to review our manuscript and the valuable suggestions. We agree that a possible triplet state should be mentioned in the manuscript and requires an additional explanation in the supplementary information (SI). We have added a sentence to the manuscript in which we mention a triplet-state structure on page 4 with reference to the SI. We have also augmented the supplementary information with a discussion paragraph of why we rule out triplet L-cystine formation.

We agree with the reviewer that a comment on the comparison of TD-DFT and ADC(2)-x calculation results is relevant. We have added a sentence on page 3.

Responses to the comments from the Reviewers 3 and 4

Reviewers' comment:

We are pleased to provide our review of the manuscript titled "UV photochemistry of the L-cystine disulfide bridge in aqueous solution investigated by femtosecond X-ray absorption spectroscopy".

The manuscript uses femtosecond X-ray absorption spectroscopy to investigate the UV photochemistry of the L-cystine disulfide bridge in aqueous solution. The authors report the homolytic cleavage of the disulfide bond upon UV irradiation, resulting in the formation of thiyl radicals as primary photoproducts. These radicals exhibit ultrafast decay, primarily due to geminate recombination, forming perthiyl radicals as secondary products. The study provides insights into the dynamic photostability of the disulfide bond and the mechanism of perthiyl radical formation, contributing to the broader understanding of disulfide photochemistry in biological contexts. We find that the investigated topic is of high relevance for the understanding of protein stability and that the study finds a novel understanding of the relevant photochemistry. We therefore recommend the manuscript for publication in Nature Communications with minor revisions as detailed below.

Reply:

We thank the reviewers 3 and 4 for their detailed report and the valuable comments. We also very much appreciate Nature Communications initiative to facilitate training in peer review of early-career researchers! In the following is our point-by-point reply.

Reviewers' comment 1:

"Thiyl radicals also do not react with water molecules and we can rule out significant formation of solvated electrons in the vicinity of thiyl radicals at our experimental peak-power

densities.” Please add a citation for this statement, as this cannot be expected to be commonly known by the broad readership of NCOMMS.

Reply 1:

This is an important point, we have added a citation at the end of the respective paragraph. We have also added a calculation in the supporting information that details the yields of two-photon excitation of water and the photogeneration of thiyl radicals with corresponding references on which we have based these calculations.

Reviewers’ comment 2:

Figure 3: The insert in Figure 3 requires a revision and a more comprehensive explanation. The insert shows delay scans at specific spectral positions associated with different photoproducts and transient loss signals. We have had some difficulty in understanding the relations between the overall figure and the inset, perhaps due to a wrong choice of colour scheme for the curves? Perhaps it just requires a better explanation, but we were not able to connect the inset with the main figure.

Reply 2:

We have addressed the deficient explanation of the inset in Figure 3 by expanding a paragraph to the manuscript in the description of Fig. 3. We have also expanded the figure caption and referred to the specific locations in the supporting information. The insert shows a 'temporal zoom' of the delay scan at 2468.2 eV. At this position, the Lorentzian absorption lineshape of the thiyl radical contributes with 8% of its peak value which is the dominant contribution at the earliest delays. At later times, the absorption of the broad shoulder between the thiyl absorption peak at 2466.8 eV and the negative bleach signal of the parent compound dominates the signal in this delay scan at 2468.2 eV. We have therefore modeled the time evolution at this energy as the sum of 8% thiyl signal and 100% secondary signal.

Reviewers’ comment 3:

Orange and Blue “Mirror Discrepancy” in Figure 3: The qualitative description that the photoproduct absorption is qualitatively mirroring the bleach signal needs clarification. It appears to refer to the observation that the absorption signal at 2466.8 eV (orange) and the bleach signal at 2472.0 eV (blue) supposedly exhibits an inverse relationship. However, this does not appear to be exactly the case at high time delays, where the photoproduct signal is mostly constant, and the bleach signal is still decaying. Please provide an explanation. A more precise explanation of this behavior in the context of the kinetic model would be beneficial.

Reply 3:

We have rephrased the paragraphs in which we discuss the data in Fig. 3 to avoid confusion. We state - as the reviewers rightfully note - that the decay of the initial product and the bleach signal are different. Indeed, the different behavior of the two time-scans in question clearly underscores that we are not observing a single product that decays back to the original substance (in which case a “mirror image” would be expected).

Reviewers' comment 4:

Also at long time delays, the fit to the photoproduct signal (orange) at 2466.8 eV requires some explanation. What is the cause of the sudden dip at 10 ps (definitely after the axis time gap)? And, given the small error bars, it seems worth discussing if there is a slow oscillation or perturbation at long time scales.

Reply 4:

The dip at 10 ps is not a real dip in the data but is due to the fact that we have introduced a break and changed from a linear to logarithmic time delay after the break. Exponential decays characteristically appear to have steps in such a log-lin plot scheme. We have added a sentence to the figure caption to clarify the lin-log transition. In the following, we plot the data between -1 ps and 50 ps with linear x-axis scaling:

We have also made an explicit reference to the supplementary section IV in which we describe the rate-equation model in full detail (first section paragraph and corresponding figure) and provide the mathematical solution to the coupled rate-equations.

Additional changes to the manuscript

We have also made changes to comply with the rules of Nature Communications, i.e.:

- shortened abstract (≤ 152 word),
- use of defined wording when referring to the supplementary information,
- simplified enumeration of chemical substances (1, 2, 3 instead of 1, I, II).

We have also corrected inconsistent numbers:

- for the threshold wavelength for C-S bond cleavage, 250nm and 200nm were mentioned previously. The correct value of 200nm is now used throughout the manuscript. Several experiments have been performed with excimer lasers with KrF gas emitting 248 nm without observation of C-S bond cleavage (which does not mean that below this wavelength C-S bond cleavage does occur). The same type of laser can be operated with ArF gas to emit 193 nm. At this wavelength C-S bond cleavage is observed.
- excitation fluence is mentioned in two places and $70\text{mJ}/\text{cm}^2$ as well as $75\text{mJ}/\text{cm}^2$ were reported previously. We have used $75\text{mJ}/\text{cm}^2$ and now report this value consistently.

Reviewer #3 (Remarks to the Author):

We are happy with the clarifications made by the authors, which have addressed most of our questions.

There is one that remains though, that the authors have not commented on: In the photoproduct signal (orange) at 2466.8 eV, it is even more apparent on the linear time scale (shown in the response), that there appears to be a semi-periodic oscillation at long delays. Were the error bars larger, they would easily be dismissed as random noise, but given the very high accuracy associated with the data points, it would be interesting if the authors would comment on this.

This should not prevent publication, but we find that it would be worthwhile if the authors would share their opinion on this observation in the paper.

Reviewer #4 (Remarks to the Author):

Responses to the comments from the Reviewers 3 and 4

Reviewers' comment:

We are happy with the clarifications made by the authors, which have addressed most of our questions.

There is one that remains though, that the authors have not commented on: In the photoproduct signal (orange) at 2466.8 eV, it is even more apparent on the linear time scale (shown in the response), that there appears to be a semi-periodic oscillation at long delays. Were the error bars larger, they would easily be dismissed as random noise, but given the very high accuracy associated with the data points, it would be interesting if the authors would comment on this.

This should not prevent publication, but we find that it would be worthwhile if the authors would share their opinion on this observation in the paper.

Reply:

We agree with the reviewers' assessment that the error bars suggest an oscillatory behavior. The error bars represent the standard deviation of the measurements at each point (600 individual measurements at 60 Hz in 10 s from which 300 difference values in fluorescence yield intensity were derived). There will be additional sources of noise that are not captured in the standard deviation of the measurement values and which manifests on timescales longer than the 10 s acquisition time for an individual data point. We have added this aspect of the data precision to the caption of Figure 3 along with information on the error bars.

From other measurements during the same beamtime we conclude that there is no significant oscillatory behavior. We have superimposed the time delay scan at 2466.8 eV with another scan that had a glitch at for about 20 seconds during that run and an additional noisy period between 1 ps and 4 ps delay:

One might discern another slight elevation at around 17 ps delay but overall, the curve is quite flat. In light of the larger variation in data points compared to an amplitude of a possible oscillatory component we do not think that we observe an oscillatory component that is statistically significant (albeit, it would be quite interesting). From a physical point of view, the period of this oscillation (~ 7 ps) would lie in the range of a few tens of wavenumbers which is too soft for an intramolecular solute mode. It would require a dephasing time constant on the very long end of typical dephasing time constants of coherent excitations in solution at ambient conditions (a few picoseconds) to observe such oscillations. But one could speculate about a mode due to the presence of a solvent cage that has previously been observed:

<https://www.science.org/doi/10.1126/science.1183799>

<https://www.nature.com/articles/s41557-020-00629-3>

Additional changes to the manuscript

- We noticed an erroneous swap in numbers for the laser repetition rates in the methods section and corrected these.
- We also noticed that unfortunately, the system of rate-equations given in the methods section stem from an older manuscript version before submission and they had not been updated. We have corrected these equations to the ones that we had actually used and described in detail in the Supplementary section. All modelling remains unchanged as do Fig. 3 and the Supplementary Information.
- The acknowledgement section was updated, two grants had not been acknowledged that funded part of the project collaboration.